# Multi-Omics Revealed the Molecular Mechanism of Maize (*Zea mays* L.) Seed Germination Regulated by GA3

**Zanping Han** *,†, **Yunqian Jin** †, **Bin Wang and Yiyang Guo**

College of Agronomy, Henan University of Science and Technology, Luoyang 471003, China;
jyq920422@163.com (Y.J.); cansong@163.com (B.W.); guoyiyang0830@163.com (Y.G.)
* Correspondence: hnlyhzp@163.com
† These authors contributed equally to this work.

**Abstract:** Maize is a valuable raw material for feed and food production. Healthy seed germination is important for improving the yield and quality of maize. However, the molecular mechanisms that regulate maize seed germination remain unclear. In this study, multi-omics was used to reveal the molecular mechanism of seed germination induced by gibberellin (GA) in maize. The results indicated that 25,603 genes were differentially expressed (DEGs) and annotated in the GO database, of which 2515 genes were annotated in the KEGG database. In addition, 791 mature miRNAs with different expression levels were identified, of which 437 were known in the miRbase database and 354 were novel miRNAs. Integrative analysis of DEGs and miRNAs suggested that carbohydrate, lipid, amino acid, and energy metabolisms are the primary metabolic pathways in maize seed germination. Interestingly, a lipid metabolism-related gene named *ZmSLP* was found to negatively regulate maize germination. We transformed this gene into *Arabidopsis thaliana* to verify its function. The results showed that the germination rate of transgenic *Arabidopsis* seeds was obviously decreased, and the growth of seedlings was weaker and slower than that of WT plants, suggesting that this gene plays an important role in promoting seed germination. These findings provide a valuable reference for further research on the mechanisms of maize seed germination.

**Keywords:** maize; seed germination; multi-omics; gibberellin; lipid metabolism

## 1. Introduction

Maize (*Zea mays* L.) is one of the most important cereal crops worldwide and is widely used as a terrestrial food, fodder, and industrial raw material [1,2]. More outstanding maize production is required to meet the growing demands of the ever-increasing world population [3]. Studies have shown that the global production of maize has recently exceeded that of rice and wheat [2]. Seed germination is a complicated process initiated by the uptake of water (imbibition) by dry seeds and ending with the emergence of the radicle through the seed coat. The following processes include the induction of translation, transcription, cell division, and energy metabolism, which involve a series of differentially expressed genes and their corresponding regulatory networks [4–7]. Therefore, the efficient and healthy germination of maize seeds, which is essential for maize production, directly affects maize yield and grain quality.

In recent decades, plant growth regulators (PGRs), including gibberellins (GA), have attracted the interest of agricultural scientists and are widely used in agronomic crops. Previous studies have reported that GAs, which are ubiquitous in higher plants, contain a large family of hormones that have long been known as endogenous GAs. GAs promote plant growth and developmental processes such as seed germination, cell division, stem elongation, dormancy, leaf expansion, flowering, and fruit development [2,8,9]. GAs and abscisic acid (ABA) are the two major phytohormones that regulate seed germination; GA promotes seed germination, whereas ABA induces seed dormancy. In addition, GA can

break quiescence and promote seed germination by increasing the content of hydrolytic enzymes, soluble sugars, and amino acids [5,10,11]. In contrast, GA-deficient mutants exhibit stronger seed dormancy and fail to complete seed germination without applying exogenous gas [12,13]. Despite extensive research being published about the molecular mechanism of maize seed germination, it remains insufficient.

Increasingly advanced technologies and research methods are used to decode the molecular mechanisms of various plant life activities, including transcriptomics, miRNA, degradome, DNA methylation sequencing, metabolomics, and isobaric tags for relative and absolute quantitation (iTRAQ) proteomic approaches. Guo et al. utilized transcription-associated metabolomics to establish a model to reveal an ABA-dependent maize acclimation mechanism to the stress combination [14]. Studies have shown that two hub genes, auxin response factor 4 (*ARF4)* and amino acid permease 3 (*AAP3)*, play central roles in the regulation of Cd-responsive genes using integration analysis of small RNAs and degradome and transcriptome sequencing in the hyperaccumulator *Sedum alfredii* [15]. High seed vigor and high-quality seed germination are important for agriculture. Therefore, to better understand the involvement and regulatory mechanism in the process, Gong et al. identified miRNAs and their targets associated with sweet corn seed vigor by combing small RNAs and degradome sequencing, finally obtaining 26 target genes cleaved by nine differentially expressed miRNAs that might play roles in the regulation of seed vigor [16]. However, reports on the molecular mechanisms regulating GAs using multi-omics research methods are rare.

In this study, two inbred maize lines, Yu537 and Yu82, were selected and treated with 400 mg/L GA3 to investigate the molecular mechanism of corn seed germination by multi-omics analysis, including transcriptome, miRNA, and degradome sequencing. This can provide valuable information for understanding the regulatory mechanism of corn seed germination.

## 2. Materials and Methods

### 2.1. Plant Material and Seed Germination

Yu82 and Yu537A, two maize (*Z. mays*) inbred lines, were selected for this study. They were obtained from the widely planted cultivar, Yuzong5. To eliminate the age and maternal effects on seeds, two inbred lines were planted in Sanya with the same management conditions. The seeds of Yu82 and Yu537A were harvested at the same developmental stage (mature stage, black layer formed). Seeds from two inbred lines were used for surface sterilization with 75% ethyl alcohol. GA3 (400 mg/L) was used to treat Yu537A seeds to promote germination. The conditions used for the germination of two seeds were below 25 °C, 14/10 h (light/dark), and an illumination intensity of 5000 lx. Purified water (20 mL) was added daily at a fixed time. Each treatment was repeated in triplicate.

### 2.2. RNA Extraction and the Library Construction of Transcriptome Sequencing

Seed embryos were cut with a sterile test blade. The fresh embryos of seeds were frozen in liquid nitrogen and stored at −80 °C. High-quality total RNA was isolated using the Tiangen RNAprep extraction kit. The concentration and quality of the RNA were checked using a Nanodrop 2000. The eligible RNAs were used for library construction and transcriptome sequencing.

### 2.3. Library Construction for Transcriptome Sequencing

Purified RNA was fragmented into short segments using a fragmentation buffer. Fragmented mRNA was used as the template to synthesize the first cDNA strand using random hexamers, and then the second was obtained using buffer solution, dNTPs, RNaseH, and DNA Polymerase I. T4 DNA polymerase and Klenow DNA polymerase were used to repair the sticky end of DNA into a flat end and add base A and adapters to the 3′ end. Finally, a sequencing library was constructed by PCR amplification. An eligible library was used

for sequencing with the Illumina Hiseq4000 platform, and the sequencing read length was double-ended at $2 \times 150$ bp (PE150).

*2.4. Filtration of Sequencing Data*

The raw data file contained short sequences (reads) of approximately 150 bp, which could not be directly used for mRNA analysis. To ensure accurate and reliable results, raw data must be preprocessed, including removing sequencing connectors (introduced during database construction) and low-quality sequencing data (due to sequencer errors).

*2.5. Comparative Analysis with the Reference Genome*

Valid data were used to map the reference genome after filtering out invalid data using the Hisat program. Gene location information specified in the genome annotation gtf file was statistically analyzed as follows: (1) read statistics of sequencing data were compared with the reference genome; (2) a summary of the regional distribution of sequencing data was compared with the reference genome; and (3) the chromosome density distribution of sequencing data was compared with the reference genome. Information regarding the regions of the reference genome can be defined as comparisons to exons, introns, and intergenic regions. Under normal circumstances, the percentage of sequence localization in the exon region should be the highest. In contrast, the comparison of reads to intron and intergenic regions may be due to splicing events of precursor mRNA, incomplete genome annotation, DNA contamination, and background noise.

*2.6. Expression Analysis of Differentially Expressed Genes*

Gene expression level was calculated with the FPKM (Fragments Per Kilobase of exon model per Million mapped reads) value. Differentially expressed genes were obtained with R-language at *p* value < 0.05 and *p* value < 0.01. The exon model FPKM value was used for gene expression measurements. The number of genes was counted in different expression regions. Based on the differentially expressed genes, GO and KEGG enrichments were analyzed in the public database.

*2.7. Association Analysis of Multi-Omics*

The miRNA sequences used in the analysis of the degradation group were all miRNAs identified by small RNA sequencing, and the database to be compared was made up of sequences spliced by transcriptome sequencing. The relationship between miRNAs and their target genes was determined using degradation group sequencing. Based on the analysis of the degradation group, we integrated the expression profiles of miRNAs and target genes in the different comparison groups to obtain an overall table of miRNAs and target genes. In addition, information was extracted from the general table to find negative regulatory relationship pairs between miRNAs and target genes in different comparison groups, and network regulatory analysis of miRNAs and target genes was conducted.

GO and KEGG enrichment analyses were performed on target genes of miRNA in each comparison group. First, the number of genes corresponding to target genes of all selected miRNAs corresponding to each function or pathway annotation was statistically calculated. A hypergeometric test was then applied to calculate the *p* value for significant enrichment. The main biological functions of the miRNA-target gene relationship can be determined by functional significance enrichment analysis, with a *p*-value $\leq 0.05$ as the threshold.

The formula for calculating the *p*-value of significance is as follows:

$$ p = 1 - \sum_{i=0}^{s-1} \frac{\binom{B}{i}\binom{TB-B}{TS-i}}{\binom{TB}{TS}} $$

where *S* is the number of annotated genes with significant expression differences in a GO item; *TS* is the number of genes with significant expression differences; *B* is the number of genes in a GO item; and *TB* is the number of total genes.

### 2.8. Acquisition of Transgenic Arabidopsis Plants

Agrobacterium vectors were inoculated into LB liquid medium containing 25 mg·L$^{-1}$ Rif and 50 mg·L$^{-1}$ kanamycin to activate the culture for 2 d. The culture conditions were set at 28 °C and 200 rpm. According to the ratio of bacteria solution to LB medium of 1:50 (*V*/*V*), 1 mL of the above bacteria solution was absorbed and added into LB liquid medium containing the corresponding resistance at 28 °C, followed by overnight culture at 200 rpm until the OD600 value was between 0.6 and 1.0. That is, the color of the medium changed from clear brick red to orange-yellow. The Agrobacterium solution with the expected OD value was centrifuged at 5000 rpm for 8 min. The supernatant was discarded to collect the bacteria. Sterile water with a 1/2 volume of bacterial solution was used to suspend the bacteria, and an equal volume of 2× suspension buffer was added before the transformation. On the second day, the mixed transformation vector was placed in a 50 mL centrifuge tube, and the inflorescences of the white flower buds were immersed in the bacterial solution for approximately 30 s. The inflorescences were then dripped onto the flower buds using a sterile dropper. After transformation, the plants were shaded in black plastic bags and placed in a paper box with 24 h of moisture. The material was placed under normal growth conditions at the end of shading. The inflorescences were retreated using the above transformation method after one week. Finally, the material was placed under normal conditions until the seeds matured. Total DNA was extracted using the improved CTAB method [17], and PCR detection was performed with specific primers for screened marker genes (*NptII*); positive plants were used for verification tests.

### 3. Results

#### 3.1. Overview of the Maize Seed Transcriptome

To investigate the molecular mechanisms of seed germination induced by GA$_3$ in maize, differentially expressed genes (DEGs) were explored using RNA-Seq technology. Six groups containing 18 samples were obtained with three replicates (Supplemental Table S1). In total, 139.54 GB of valid bases with a mean GC content of 53.08% were obtained. After data processing, the Q30 values ranged from 91.69% to 95.49% (Supplemental Tables S2 and S3). In total, 138,049 transcripts and 44,117 genes with different expression levels were identified. Among these, 25,603 genes were annotated in the gene ontology (GO) database, and 2515 genes were annotated in the KEGG (Kyoto Encyclopedia of Genes and Genomes) database. All valid reads were mapped to the reference genome (maize B73) (Table 1). The average mapped ratio was 87.53%, providing a reliable reference. According to the genomic region information of the reference genome, the mapping regions could be divided into exon, intron, and intergenic regions. The results indicated that the percentage of exon regions was the highest. The percentage of intron regions was the lowest (Supplemental Figure S1), which may correlate with the cleavage events of pre-mRNAs, incomplete genome annotation, DNA contamination, and background noise.

**Table 1.** Statistical analysis of mapped reads.

| Sample | Valid Reads | Mapped Reads | Unique Mapped Reads | Multi Mapped Reads | PE Mapped Reads | Non-Splice Reads | Splice Reads |
|---|---|---|---|---|---|---|---|
| E_6h1 | 44,522,844 | 38,872,898 (87.31%) | 25,118,045 (56.42%) | 13,754,853 (30.89%) | 35,328,974 (79.35%) | 23,649,652 (53.12%) | 1,049,188 2(23.57%) |
| E_6h2 | 52,779,658 | 45,489,407 (86.19%) | 27,416,327 (51.94%) | 18,073,080 (34.24%) | 40,859,244 (77.41%) | 26,677,569 (50.55%) | 12,594,962 (23.86%) |
| E_6h3 | 48,573,036 | 42,144,424 (86.77%) | 25,152,219 (51.78%) | 16,992,205 (34.98%) | 38,151,374 (78.54%) | 24,484,992 (50.41%) | 11,414,228 (23.50%) |
| S_6h1 | 56,094,828 | 47,116,337 (83.99%) | 26,205,903 (46.72%) | 20,910,434 (37.28%) | 41,867,540 (74.64%) | 26,352,750 (46.98%) | 12,264,970 (21.86%) |
| S_6h2 | 56,605,766 | 47,223,659 (83.43%) | 27,515,806 (48.61%) | 19,707,853 (34.82%) | 41,761,020 73.78%) | 28,352,616 (50.09%) | 12,250,405 (21.64%) |
| S_6h3 | 56,941,752 | 47,673,165 (83.72%) | 28,036,288 (49.24%) | 19,636,877 (34.49%) | 42,222,202 (74.15%) | 28,223,671 (49.57%) | 12,889,242 (22.64%) |
| E_53h1 | 51,623,746 | 45,966,061 (89.04%) | 29,463,816 (57.07%) | 16,502,245 (31.97%) | 42,035,450 (81.43%) | 22,812,230 (44.19%) | 18,799,134 (36.42%) |
| E_53h2 | 51,263,292 | 45,899,826 (89.54%) | 31,031,019 (60.53%) | 14,868,807 (29.00%) | 42,108,328 (82.14%) | 23,255,978 (45.37%) | 19,879,897 (38.78%) |
| E_53h3 | 56,832,242 | 50,841,433 (89.46%) | 32,269,169 (56.78%) | 18,572,264 (32.68%) | 46,704,020 (82.18%) | 26,033,538 (45.81%) | 19,548,995 (34.40%) |
| S_53h1 | 54,294,104 | 47,469,186 (87.43%) | 30,395,012 (55.98%) | 17,074,174 (31.45%) | 43,186,082 (79.54%) | 23,781,631 (43.80%) | 19,888,369 (36.63%) |
| S_53h2 | 48,938,972 | 43,265,705 (88.41%) | 27,044,065 (55.26%) | 16,221,640 (33.15%) | 39,668,690 (81.06%) | 21,719,154 (44.38%) | 16,712,438 (34.15%) |
| S_53h3 | 50,377,860 | 44,288,776 (87.91%) | 28,249,543 (56.08%) | 16,039,233 (31.84%) | 40,480,912 (80.35%) | 22,362,326 (44.39%) | 18,059,268 (35.85%) |
| SGA_53h1 | 53,650,262 | 47,512,991 (88.56%) | 30,986,382 (57.76%) | 16,526,609 (30.80%) | 43,394,170 (80.88%) | 23,359,239 (43.54%) | 20,725,004 (38.63%) |
| SGA_53h2 | 45,060,898 | 39,937,286 (88.63%) | 24,497,940 (54.37%) | 15,439,346 (34.26%) | 36,729,340 (81.51%) | 20,075,697 (44.55%) | 15,951,613 (35.40%) |
| SGA_53h3 | 55,721,064 | 49,400,868 (88.66%) | 30,618,696 (54.95%) | 18,782,172 (33.71%) | 45,147,652 (81.02%) | 24,821,328 (44.55%) | 20,102,437 (36.08%) |
| S_78h1 | 51,162,146 | 44,975,042 (87.91%) | 28,480,225 (55.67%) | 16,494,817 (32.24%) | 41,083,328 (80.30%) | 23,069,493 (45.09%) | 18,061,692 (35.30%) |
| S_78h2 | 53,589,910 | 48,079,252 (89.72%) | 31,263,293 (58.34%) | 16,815,959 (31.38%) | 44,274,240 (82.62%) | 23,985,179 (44.76%) | 21,043,894 (39.27%) |
| S_78h3 | 42,286,200 | 37,561,743 (88.83%) | 23,876,043 (56.46%) | 13,685,700 (32.36%) | 34,416,830 (81.39%) | 18,772,494 (44.39%) | 16,135,439 (38.16%) |

Note: Samples and sample names for sequencing. Valid reads represent clean data obtained after quality control. The mapped reads represent the number of reads matched to the genome. Uniquely Mapped reads represent the number of reads that can be uniquely matched to one location in the genome. Multi-mapped reads represent the number of reads that can be mapped to multiple locations in a genome. PE Mapped reads are paired-end sequencing reads mapped to the genome. Reads mapped to the sense strand represent the reads that were mapped to the sense strand. Reads mapped to the antisense strand represent reads that were mapped to the antisense strand. Non-splice reads represent reads that can be mapped to genomic regions by end-to-end alignment. Splice reads cannot be mapped to genomic regions by end-to-end alignment.

### 3.2. Analysis of DEGs Induced by GA3 in the Germination Process of Maize Seeds

The fragments per kilobase of transcript per million mapped reads (FPKM) value was used to represent the expression of each gene (Supplemental Figure S2), and significant DEGs were screened based on a *p*-value < 0.05. The number of significant DEGs in the

different groups was counted (Figure 1), and the results showed noticeable expression changes. A total of 4312 significant DEGs were induced in the E_53h vs. E_6h group, of which 2593 DEGs were upregulated and 1719 DEGs were downregulated. A total of 5696 significant DEGs were identified in the S_53h vs. S_6h group, of which 3760 were upregulated and 1936 were downregulated. In the group of SGA_53h vs. S_6h, a total of 4815 DEGs were discovered, including 2104 upregulated DEGs and 2711 downregulated DEGs. Only 2704, 1086, and 835 DEGs were obtained from groups S_78h vs. S_6h, S_78h vs. SGA_53h, and SGA_53h vs. S_53h, respectively. Drastically reduced numbers of DEGs in the later stages of maize seed germination suggest that the regulation of the early stages of imbibition germination of maize seeds is more complex and requires more DEGs. After GA3 treatment, only 835 significant DEGs were induced in the SGA_53h sample compared to the S_53h sample, of which 322 were upregulated and 513 were downregulated, indicating that GA3 may induce other specific germination-related genes.

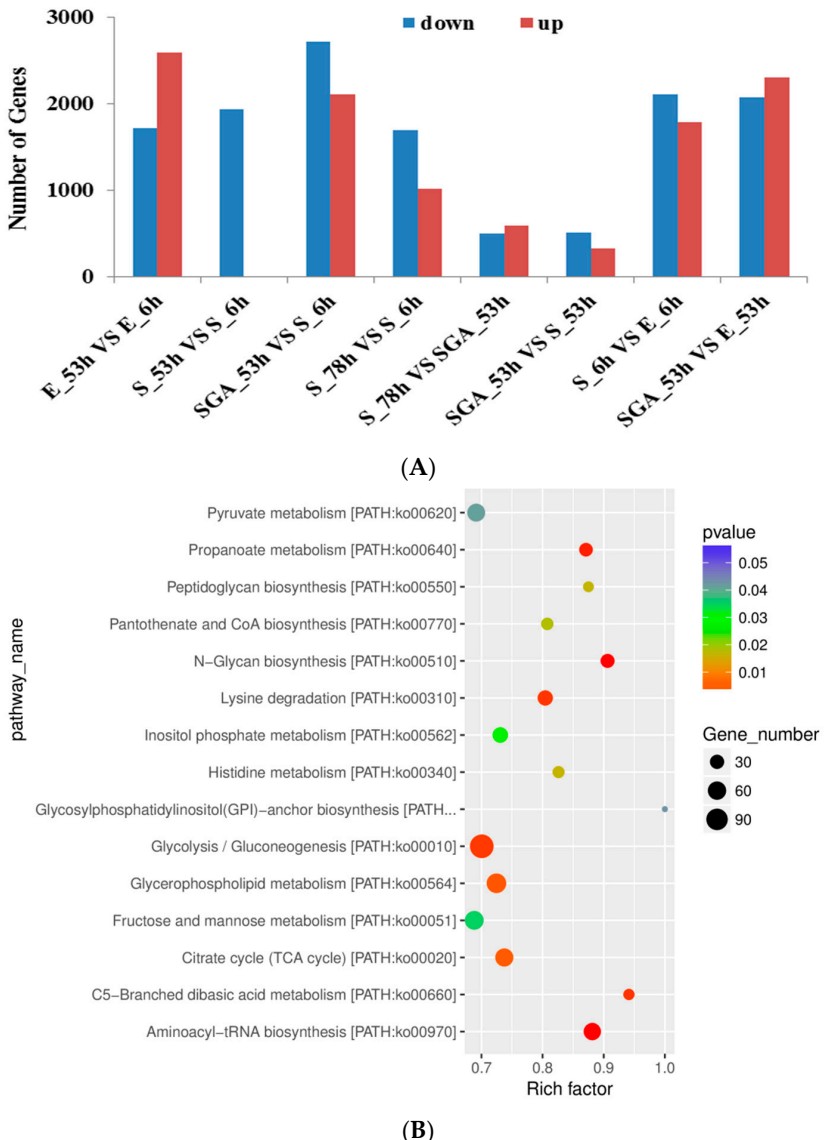

(**A**)

(**B**)

**Figure 1.** Number of significant differentially expressed genes (DEGs) in different groups and KEGG analysis of DEGs. (**A**) Number of significant DEGs in different groups. Red and blue indicate upregulation and downregulation, respectively. (**B**) KEGG analysis of significant DEGs. Different colors represent different *p*-values, and the size of the color dot represents the number of DEGs; the bigger the dot, the more enriched the genes.

GO analysis (including biological processes, cellular components, and molecular functions) was performed to analyze the functions of the DEGs. The results showed that oxidation-reduction (GO: 0055114), carbohydrate metabolic (GO: 0005975), integral components of the membrane (GO: 0016021), mitochondrion (GO: 0005739), ATP binding (GO: 0005524), protein binding (GO: 0051082), and catalytic activity (GO: 0003824) processes were the most enriched terms. KEGG analysis showed that the significant DEGs were mainly enriched in Gly/gluconeogenesis (ko00010), glycerophospholipid (ko00564), fructose and mannose (ko00051), and pyruvate metabolism (ko00620). To further determine the accuracy of the RNA sequencing results, 15 DEGs involved in the regulation of seed germination induced by GA3 were selected for RT-qPCR, and specific primers for these genes were designed using online NCBI primer software. RNA sequencing results showed that the expression levels of eight DEGs were significantly upregulated and seven DEGs were significantly downregulated. The RT-qPCR results showed that the expression levels of the selected DEGs were consistent with the RNA-seq data, which proved that the transcriptome sequencing data of corn seeds were reliable.

### 3.3. miRNAs Were Involved in the Germination Regulation of Maize Seeds

miRNAs have been reported to be involved in the regulation of corn seed vigor. Several miRNAs with specific functions have been reported in maize seeds [16,18,19]. To fully understand the regulatory mechanism of maize seeds, miRNA sequencing was performed. Significant DEGs in the germination process of maize seeds were explored. In total, 791 mature miRNAs with different expression levels were identified (Supplemental Figure S3). Of these, 437 were already known in the miRbase database, and 354 were novel miRNAs. As shown in Figure 2, compared with 6 HAI, 28 miRNAs with different expression levels were obtained at 53 HAI ($p < 0.05$) in Yu82, of which 15 miRNAs were upregulated and 13 were downregulated. Only six miRNAs were upregulated at 53 HAI in Yu537A ($p < 0.05$), indicating that the regulatory mechanisms in Yu82 were more complicated than those in Yu537A. After GA3 treatment, the number of differentially expressed miRNAs increased to 22 from six, suggesting that GA3 induced more DEGs involved in maize seed germination. Among the 22 significantly differentially expressed miRNAs, only one was downregulated, whereas the others were upregulated. Genome location clusters of pre-miRNAs showed that approximately 26.00% were located on chromosome 2, indicating that the miRNAs on chromosome 2 played an important role in maize seed germination.

Mature plant miRNAs generally interact with their targets through perfect or near-perfect complementarity, leading to the cleavage of target mRNAs [20–22], which is guided by the RNA-induced silencing complex (RISC) [23,24]. In our study, all the miRNA targets were identified using prediction and degradome sequencing. To identify the effective targets of the miRNAs, we investigated all the targets of significantly differentially expressed miRNAs in each comparison group. In the E_53h vs. E_6h group, 28 significantly differentially expressed miRNAs and 113 target genes were gained with negative expression. While only four significantly differentially expressed miRNAs with 22 target genes were observed to have negative expressions. However, after the GA3 treatment, the number of significantly differentially expressed miRNAs, and their target genes increased to 14 and 41, respectively. These results indicate that GA3 induced significantly more differentially expressed miRNAs to regulate germination-related genes and promote the germination of maize seeds.

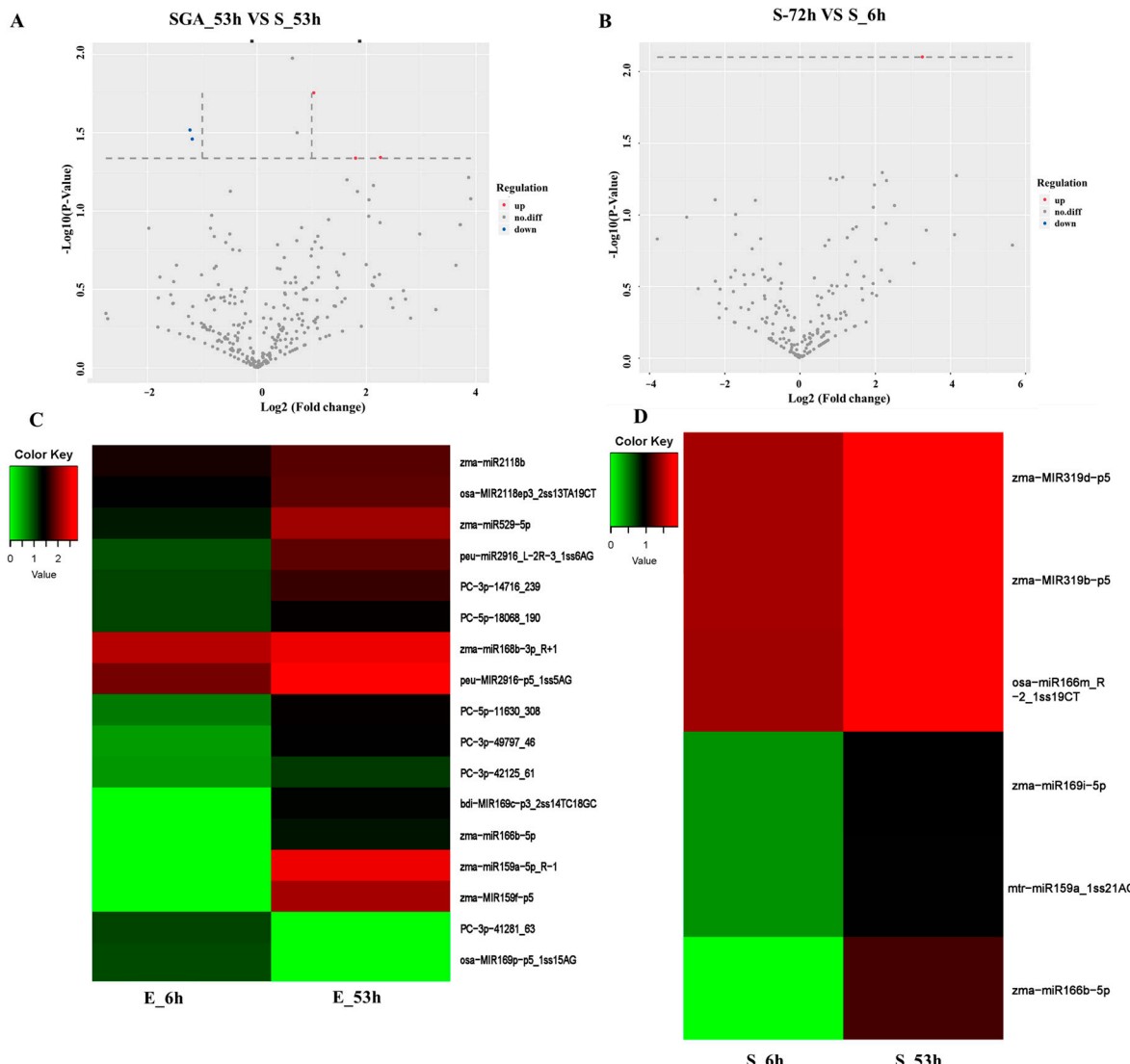

**Figure 2.** Differentially expressed miRNAs in two different germination stages in two maize inbreds. (**A**) Volcano of differentially expressed miRNAs in the SGA_53h vs. S_53h. (**B**) Volcano of differentially expressed miRNAs in the group of S-72h vs. S_6h. (**C**) Heatmap analysis of differentially expressed miRNAs in the E_53h vs. E_6h. (**D**) Heatmap analysis of differentially expressed miRNAs in the SGA_53h vs. S_53h.

*3.4. Integrative Analysis of DEGs and miRNAs Involved in the Germination of Maize Seed Treated with GA3*

To further understand the molecular mechanism of increased maize seed vigor induced by GA3, multi-omics, including transcriptome, miRNA, and degradome sequencing, were used to identify key genes related to maize seed vigor. The results showed that in the SGA_53h vs. S_53h group, 43 miRNA-mRNA pairs were identified at a significance level of $p \leq 0.05$. As shown in Figure 3A, GO enrichment analysis of the differentially expressed miRNAs and DEGs revealed that protein binding (GO: 0005515), ATP binding (GO: 0005524), and lipid metabolic processes (GO: 0006629) were the most enriched. In addition, 355 and 1781 miRNA-mRNA pairs were identified in S_53h vs. S_6h and E_53h vs. E_6h, respectively. GO enrichment analysis showed these differentially expressed miRNA-mRNA pairs were mainly enriched in items of oxidation-reduction (GO: 0055114), carbohydrate binding (GO: 0030246), hydrolase activity (GO: 0016787), zinc ion binding (GO: 0008270), porphyrin-containing compound biosynthesis (GO: 0006779), protein glycosylation (GO: 0006486), and ATP binding (GO: 0005524) processes (Figure 3A). Furthermore, ATP binding

(GO: 0005524) and lipid metabolic processes (GO: 0006629) were enriched in S_53h vs. S_6h and E_53h vs. E_6h, indicating DEGs in these items play a vital role in the energy supply during germination. KEGG enrichment analysis showed that carbohydrate, lipid, amino acid, and energy metabolisms were the main metabolic pathways involved in maize seed germination (Figure 3B).

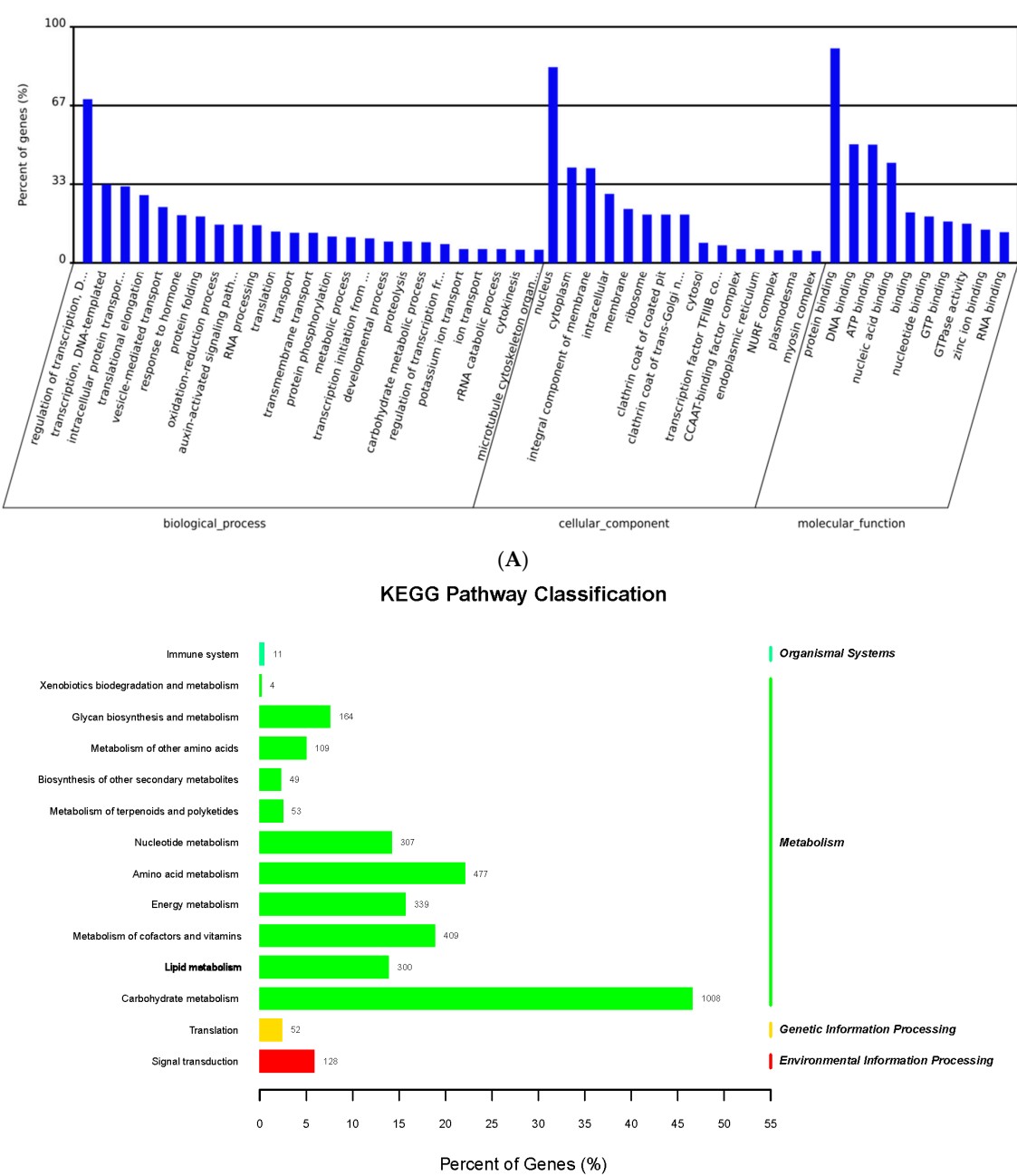

**Figure 3.** GO and KEGG enrichment analysis of miRNA-mRNA targets. (**A**) GO enrichment analysis of miRNA-mRNA targets. (**B**) KEGG enrichment analysis of miRNA-mRNA targets.

Therefore, it is speculated that the primary seed germination process mainly uses the energy stored in the seed body to activate the corresponding hydrolase and protease activities and redox reactions to hydrolyze sugar substances to release energy and promote germination. Maize is an important food and feed crop, and the hydrolysis of lipid metabolism negatively regulates the germination of maize seeds. Therefore, differentially

expressed miRNAs and mRNAs in lipid metabolic processes are crucial for the healthy germination of maize seeds.

### 3.5. Function Verification of the Lipid Metabolism-Related Gene ZmSLP

To explore the molecular mechanism regulating maize seed vigor, expression profiles of differentially expressed miRNAs and mRNAs were analyzed, and finally the target gene LOC100282421 was obtained from the differential metabolic pathway K00561 (glycerolipid metabolism) and named *ZmSLP* (Figure 4A–D). A miRNA–mRNA pair, zma-MIR169c-3p–ZmSLP, was identified and is involved in seed germination regulation. Under normal conditions, low expression of the miRNA zma-MIR169c-3p induces high expression of the target gene *ZmSLP*. While at the germination stage, the miRNA zma-MIR169c-3p was upregulated to inhibit the expression of the target gene *ZmSLP*, promoting seed germination. To verify the function of *ZmSLP* in regulating maize seed vigor, an expression vector for *ZmSLP* was constructed and transfected into *Arabidopsis thaliana*. The results indicated that transgenic *A. thaliana* seeds germinated earlier and the seedlings were more robust than wild-type *A. thaliana* plants (Figure 4E,F), suggesting that *ZmSLP* significantly improved the germination ability of transgenic *A. thaliana* seeds.

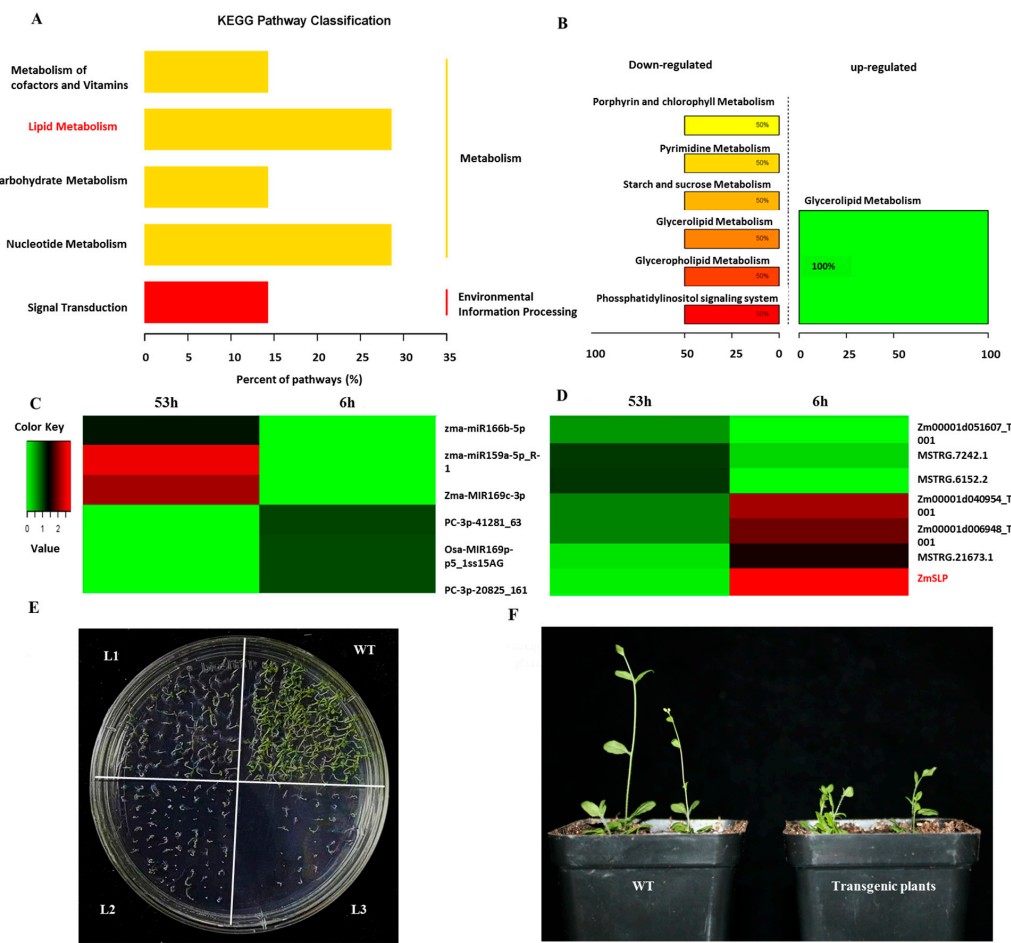

**Figure 4.** KEGG enrichment and expression analysis of target *ZmSLP*. (**A**) Classification of significantly enriched KEGG pathways. (**B**) Ratio of upregulated or downregulated genes in significantly enriched pathways. (**C**) Expression analysis of partial miRNAs. (**D**) Expression analysis of the target *ZmSLP* gene. (**E**) Comparison of Arabidopsis seed germination on solid medium. WT represents wild-type Arabidopsis seeds. L1, L2, and L3 represent three transgenic Arabidopsis lines. (**F**) Growth difference between WT and transgenic plants.

## 4. Discussion

Maize (*Zea mays* L.) is an excellent model plant for genetic studies and an important staple crop typically used as a raw material for human food, animal feed, and industrial production [25]. Healthy seed germination is an important event for high-yield and high-quality maize and involves a series of physiological and genetic regulations, such as miRNAs, an important class of non-coding RNAs [26]. GA is generally regarded as a phytohormone that releases seed dormancy and promotes seed germination [27]. However, ABA is another important phytohormone that regulates seed germination, and the GA-ABA ratio is a key factor in breaking and promoting seed germination [28]. Several thousand DEGs were identified in different comparison groups, and the early stages of imbibition germination of maize induced more DEGs. GO and KEGG enrichment analysis suggested that multiple items were closely correlated with the seed germination process, including the oxidation-reduction process (GO: 0055114), carbohydrate metabolic process (GO: 0005975), an integral component of membrane (GO: 0016021), mitochondrion (GO: 0005739), ATP binding (GO: 0005524), Gly/gluconeogenesis (ko00010), glycerophospholipid (ko00564), fructose and mannose (ko00051), and pyruvate metabolism (ko00620). Among these, using informatics analysis, the lipid metabolism-related gene *ZmSLP* was identified for its close correlation with maize seed germination.

Oil bodies are carriers of lipids in maize seeds and provide energy for seed germination and early seedling growth. Lipids exist in different species as triacylglycerols (TAG) [29–31]. As an excellent food crop with high nutritional value, maize is an important raw material source for feed for the animal husbandry, aquaculture, and fish industries. In this study, glycerophospholipid metabolism (ko00564) was significantly enriched, suggesting that lipid metabolism plays an important role in maize seed germination. Furthermore, the *ZmSLP* gene was identified and functionally verified in transgenic *Arabidopsis* plants, and the overexpression of this gene significantly reduced the germination quality. The main nutrients and energy used for seed germination are mostly derived from stored starch, because maize is also one of the most important food crops in our country due to its high content of starch. At the seed germination stage, starch provides energy and a carbon skeleton for seed germination, which is necessary for maize seed germination. Li et al. discovered a putative rice non-specific lipid transport protein named *OsLTPL23*, which plays an important role in the seed germination process [32]. Hydro-electro hybrid priming (HEHP), a new seed introduction technique, promotes carrot seed germination by activating lipid metabolism [33].

In the process of maize seed germination, free amino acids and soluble sugars in the embryo are first consumed, and the utilization of nutrients in the storage tissue begins only when the seeds absorb water fully and expand to a certain stage. Overexpression of *ZmSLP* in Arabidopsis would lead to a disorder of lipid metabolism, inhibiting the germination of maize seeds and strong growth at the seedling stage. In this study, we observed that lipid metabolism is important for seed germination. Therefore, we established a regulation model for maize seed germination, providing valuable information for molecular research on maize seed germination.

**Supplementary Materials:** The following supporting information can be downloaded at: https://www.mdpi.com/article/10.3390/agronomy13071929/s1, Figure S1: Distribution regions of mapped reads in each sample; Figure S2: Expression Profiles of genes obtained in each sample; Figure S3: Differentially expressed of partial miRNAs among different samples; Table S1: Information of each sample; Table S2: Quality of RNA sequencing reads of maize seeds; Table S3: Statistical results of the comparisons with reference genome.

**Author Contributions:** Z.H. and Y.J. designed the experiments and wrote the manuscript. B.W. and Y.G. helped with the experiments and manuscript revisions. All authors have read and agreed to the published version of the manuscript.

**Funding:** This study was supported by grants from the National Natural Science Foundation of China (No. U1504315) and the Science and Technology Project in Henan Province (No. 212102110244).

**Institutional Review Board Statement:** Not applicable.

**Informed Consent Statement:** Not applicable.

**Data Availability Statement:** The datasets generated in this study are available from the GEO repository (accession number: GSE196738). All analysis results generated during this study are included in this article and the Supplementary Information.

**Acknowledgments:** We thank Hangzhou LC-Bio Technology Co., Ltd. for their assistance with sequencing and bioinformatics analysis.

**Conflicts of Interest:** The authors declare no conflict of interest.

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
