# Peer review of "Multi-Omics Revealed the Molecular Mechanism of Maize (Zea mays L.) Seed Germination Regulated by GA3"

_agronomy, doi:10.3390/agronomy13071929_

Round 1

Reviewer 1 Report

There are some observations in the manuscript that should be addressed by the authors.

Author Response

Dear editor and reviewer 1, 

   Thanks for your patience and valuable comments. Now we have revised the manuscript  according the comments point-by-point.  

Point 1: Drastically, reduced .....

Response 1: We have revised it in the manuscript. 

Point 2: Words are unclear

Response 2: We have replaced it with a new picture in the manuscript. 

Point 3: were used

Response 3: We have revised it in the manuscript. 

Point 4: were used

Response 4: We have revised it in the manuscript. 

Point 4: No, maize is not an oil crop.

Response 4: We have revised it in the manuscript. 

Point 5: Move this section before results.

Response 5: We have moved the "Materials and methods“ section before results in the manuscript. In addition, we also revised the reference number. 

Reviewer 2 Report

This research is about molecular mechanisms that regulate maize seed germination. Multi-omics was used to reveal the molecular mechanism of seed germination induced by gibberellin (GA) in maize.

They found interesting result that lipid metabolism plays an important role in maize seed germination. This research could provide valuable information for molecular research on maize seed germination. The influence of GA on the promotion of germination can find its practical application in corn seed production. Therefore, knowledge of the mechanisms of GA influence on maize seed germination is important. It would be interesting to examine whether the application of GA can have a positive effect on the germination of maize seeds with impaired germination. If there is such information, supplement the discussion with it.

I think that the manuscript can be published with minor corrections. Some comments are given in the text.

Author Response

Dear editor and reviewer 2, 

     Thanks for your patience and valuable comments. Now we have revised the manuscript according the comments point-by-point. 

Point 1: put the serial number of the reference - 16?

Response 1: We have revised it in the manuscript. 

Point 2: delete: (Gong, Ding, et al., 2015)(16)

Response 2: We have revised it in the manuscript. 

Point 3: arrange the table 1

Response 3: We have rearranged it in the manuscript. 

Point 4: full name if mentioned for the first time

Response 4: We have added full name information in the manuscript. 

Point 5: It is not marked in the picture (A) i (B)

Response 5: We have added figure 3B in the manuscript. 

Point 6: It is not clear where this can be seen on the graph 3B.

Response 6: We have added figure 3B in the manuscript. 

Point 7: It is not clear where this can be seen on the graph 3B.

Response 7: We have added figure 3B in the manuscript. 

Point 8: target

Response 8: We have revised it in the manuscript. 

Point 9: Improve the quality of A-D charts. The text is unclear on them.

Response 9: We have improved the figure quality in the manuscript. 

Reviewer 3 Report

The MS entitled; Multi-omics established the molecular mechanism of Zea mays seed germination is very poor, and I recommend major revision to improve MS quality.

·         The introduction needs to be to the point and revise it.

·         Table 1 is hard to read, I recommend to minimize his size or put as landscape.

·          Fig. 1, very poor quality, must change it

·         All figure are very poor quality, need to add original figures.

·         Need to improve the discussion significantly, and explain why arabidopsis seed germinations are effected? What is this line, and this is OE or mutants?. And explain what kind of gene this is?

·         I didn’t find any solid reason or explanation in dissection section.   

·         Put full detail in figure 4 caption. To what’s you add.

·         Material and method; add full information about line you used and location where you did experiment.

·         I recommend author to revise this MS carefully, and need to add more information.

must be improved

Author Response

Dear editor and reviewer 3, 

   Thanks for your patience and valuable comments. Now we have revised the manuscript according to the comments point-by-point. 

Point 1: The introduction needs to be to the point and revise it.

Response 1: We have revised the title of the manuscript. In troduction section, we introduced the research progress about GA regulation, multi-omics and the aim of our research. 

Point 2: Table 1 is hard to read, I recommend to minimize his size or put as landscape.

Response 2: We have revised the table 1. 

Point 3: Fig. 1, very poor quality, must change it

Response 3: We have revised the figure 1.

Point 4: All figure are very poor quality, need to add original figures.

Response 4: We have revised all figures, including figure 2, figure 3 and figure 4. 

Point 5: Need to improve the discussion significantly, and explain why arabidopsis seed germinations are effected? What is this line, and this is OE or mutants?. And explain what kind of gene this is?

Response 5: We have added more information in discussion. 

Point 6:   I didn’t find any solid reason or explanation in dissection section. 

Response 6:  We have added descriptions about why transgenic seeds germinate slowly.

Point 7:   Put full detail in figure 4 caption. To what’s you add.

Response 7:  We have added more details in figure 4 caption. 

Point 8:  Material and method; add full information about line you used and location where you did experiment.

Response 8:  We have added more details in "Material and method" section .